# Purification Efficiency of Three Combinations of Native Aquatic Macrophytes in Artificial Wastewater in Autumn

**DOI:** 10.3390/ijerph18116162

**Published:** 2021-06-07

**Authors:** Lei Xu, Siyu Chen, Ping Zhuang, Dongsheng Xie, Xiaoling Yu, Dongming Liu, Zhian Li, Xinsheng Qin, Faguo Wang, Fuwu Xing

**Affiliations:** 1Key Laboratory of Plant Resources Conservation and Sustainable Utilization, Guangdong Provincial Key Laboratory of Applied Botany, South China Botanical Garden, Chinese Academy of Sciences, Guangzhou 510650, China; xulei@scbg.ac.cn (L.X.); xdsdyxhh0@163.com (D.X.); yuxl@scib.ac.cn (X.Y.); liudm@scib.ac.cn (D.L.); xinfw@scbg.ac.cn (F.X.); 2School of Marine Sciences, Sun Yat-sen University, Guangzhou 510275, China; chensy540@163.com; 3Southern Marine Science and Engineering Guangdong Laboratory (Guangzhou), Guangzhou 511458, China; zhuangp@scbg.ac.cn (P.Z.); lizan@scbg.ac.cn (Z.L.); 4College of Life Sciences, Zhongkai University of Agriculture and Engineering, Guangzhou 510225, China; 5College of Forestry and Landscape Architecture, South China Agricultural University, Guangzhou 510642, China

**Keywords:** aquatic plants, nitrogen and phosphorus, purification efficiency

## Abstract

Water pollution caused by excessive nutrient and biological invasion is increasingly widespread in China, which can lead to problems with drinking water as well as serious damage to the ecosystem if not be properly treated. Aquatic plant restoration (phytoremediation) has become a promising and increasingly popular solution. In this study, eight native species of low-temperature-tolerant aquatic macrophytes were chosen to construct three combinations of aquatic macrophytes to study their purification efficiency on eutrophic water in large open tanks during autumn in Guangzhou City. The total nitrogen (TN) removal rates of group A (*Vallisneria natans* + *Ludwigia adscendens* + *Monochoria vaginalis* + *Saururus chinensis*), group B (*V. natans* + *Ipomoea aquatica* + *Acorus calamus* + *Typha orientalis*), and group C (*V. natans + L. adscendens + Schoenoplectus juncoides + T. orientalis*) were 79.10%, 46.39%, and 67.46%, respectively. The total phosphorus (TP) removal rates were 89.39%, 88.37%, and 91.96% in groups A, B, and C, respectively, while the chemical oxygen demand (COD) removal rates were 93.91%, 96.48%, and 92.78%, respectively. In the control group (CK), the removal rates of TN, TP, and COD were 70.42%, 86.59%, and 87.94%, respectively. The overall removal rates of TN, TP, and COD in the plant groups were only slightly higher than that in CK group, which did not show a significant advantage. This may be related to the leaf decay of some aquatic plants during the experiment, whereby the decay of *V. natans* was the most obvious. The results suggest that a proper amount of plant residue will not lead to a significant deterioration of water quality.

## 1. Introduction

The pollution of water sources has been an issue of considerable public interest over the past few decades [1]. The wastewater pollution by excessive nitrogen (N) and phosphorus (P) has become a widespread global problem, leading to the degradation of aquatic ecosystems, a decline in biodiversity, the collapse of nutrient cycles, and the development of water blooms [2,3,4]. For example, eutrophication has caused negative effects on agricultural, industrial, and drinking water production [5]. To address these serious issues, various physical, chemical, and biological measures have been applied to remediate N and P pollution over the past several decades [6,7,8], including aeration, water diversion, sediment dredging, chemical flocculation, chemical algaecide addition, and in situ chemical reaction techniques. Some of these approaches have high environmental costs because they can cause secondary pollution for the use of chemical agents, which can even destroy river ecosystems [9,10]. In recent years, phytoremediation using aquatic plants with high productivities and nutrient removal capabilities has received increasing public attention [11,12]. Phytoremediation uses plants to mitigate, transfer, stabilize, or degrade pollutants in soil, sediment, and water. This technique has been extensively applied in rivers [13], lakes [14], and constructed wetlands [15,16].

Vegetation is an important component that plays a key role in aquatic environments. As primary producers, plants supply food to the first consumers in trophic chains [17]. Vegetation also provides habitats and refuges for periphyton [18], zooplankton [19], and other invertebrate species [20]. Wastewater treatment in aquatic macrophyte systems occurs through several mechanisms, including solid settling, plant uptake of contaminants, biotransformation, and physicochemical reactions [12]. Previous studies on the phytoremediation of eutrophic waterbodies were usually carried out during warm seasons and mainly focused on the effect of individual plant species [21,22,23]. A study reported that water hyacinth achieved the highest removal efficiency for total N (TN) (89.4%) and ammonium N (NH_4_^+^-N) (99.0%) during a static experiment under a water temperature of 28~36 °C, while water lettuce exhibited the highest removal efficiency for total P (TP) (93.6%) [8]. However, fewer studies have reported on their efficacy in autumn and winter. There are insufficient systematic studies on the annual efficiency of water purification by aquatic plants. Therefore, it is meaningful to do some research on phytoremediation in low temperature season.

In fact, the nutrient absorption and water purification effects of different plants vary greatly, and the same plant may have a better removal effect on one index of water quality, while the effect on other indexes may be relatively poor. Single plant systems are temporally and spatially limited in their ability to assimilate N and P, thus potentially affecting the N and P removal rates. On the other hand, multi-species systems should deliver good performance by buffering against variations in weather and nutrient conditions as a result of their diversity and adaptability [24]. Some studies have shown that multiple plants can purify water quality better than a single plant owing to a more reasonable species diversity, which makes it easier to maintain the long-term stability of the ecological system [5,25]. Coleman et al. [26] found that the mixed planting of *Schoenoplectus tabernaemontani*, *Juncus effusus*, and *Typha latifolia* had a better purification effect on domestic sewage than a single species. However, some studies have pointed out that there is no significant difference between mixed and single planting [27]. The removal mechanisms and effects of water nutrients using combinations of different aquatic plants need to be explored more effectively.

Plant invaders can greatly diminish the abundance or survival of native species and can completely alter the native ecosystem in terrestrial and freshwater habitats [28]; however, exotic plants with strong reproduction are widely used in China [29]. For instance, *Eichhornia crassipes* may be the most important and commonly used species of aquatic macrophytes used to treat wastewater [30].

The aim of this study is to compare the potential of native aquatic plant combinations for the restoration of eutrophic waterbodies during autumn. In this study, various combinations of selected plant species were tested for their ability to remove nutrients from wastewater in open tanks over 54 days. The results provide a reference for engineering practice of water restoration in subtropical areas during autumn.

## 2. Materials and Methods

### 2.1. Plant Species

The plants used in this experiment were listed in Table 1 and were selected from the countryside in the Guangdong Province. All plants were grown in running water for a week before starting the experiment. The stem of *Saururus chinensis* was placed directly on the water surface to make it grow freely like a floating plant, while *Monochoria vaginalis* was fixed on the water surface with a container like the plants in an ecological floating bed. The roots of other plants were anchored to the bottom with sand.

### 2.2. Wastewater Preparation

Many previous studies have directly added nutrients to tap water or distilled water to produce artificial wastewater, thus ignoring the synergistic effect of microorganisms and plants in the water purification process [5,31]. In order to simulate natural eutrophic water as much as possible, the artificial wastewater in this experiment was prepared using C_6_H_12_O_6_·H_2_O (207 mg/L), NH_4_Cl (38 mg/L), and KH_2_PO_4_ (8.7 mg/L) dissolved in natural wastewater with a low nutrient content to prepare the designed average concentrations as follows: chemical oxygen demand (COD) of 207 mg/L, TN of 10 mg/L, and TP of 2 mg/L, respectively. In all cases, we used a trace mineral solution (1 mL/L) containing EDTA-Na (0.1 g/L), FeSO_4_·7H_2_O (0.1 g/L), MnCl_2_·4H_2_O (0.1 g/L), CoCl_2_·6H_2_O (0.184 g/L), CaCl_2_ (0.05 g/L), ZnCl_2_ (0.1 g/L), CuCl_2_·2H_2_O (0.015 g/L), NiCl_2_·6H_2_O (0.03 g/L), H_3_BO_3_ (0.01 g/L), Na_2_MoO_4_·2H_2_O (0.01 g/L), and H_2_SeO_3_ (0.001 g/L) [5,16,32].

The initial water quality parameters are as follows: temperature (T, 23.26 ± 0.25 °C); pH (7.41 ± 0.14); dissolved oxygen (DO, 1.29 ± 0.95 mg/L); TN (9.70 ± 1.51 mg/L); TP (1.81 ± 0.23 mg/L); COD (343 ± 133 mg/L).

### 2.3. Experimental Design

A total of 12 rectangular open plastic tanks, measuring 150 cm in length, 150 cm in width, and 50 cm in height, were used for the experiments with 576 L of artificial wastewater and 800 g of aquatic plants. The tanks were divided into four groups: (A) *V. natans + L. adscendens + M. vaginalis + S. chinensis*; (B) *V. natans + I. aquatica + A. calamus + T. orientalis*; (C) *V. natans + L. adscendens + S. juncoides + T. orientalis*; and (CK) control (no plants). Each group contained three tanks.

Water samples were collected on days 0, 3, 8, 16, 27, and 54 by dipping a 300 mL graduated cylinder at three locations across the container surface during the mid-morning and then combining these. The aliquots of each sample were filtered, and both the filtered and unfiltered portions were immediately stored at 4 °C. The parameters measured included T, pH, DO, TN, TP, and COD. Losses in the culture volume due to evapotranspiration were countered by the addition of tap water to the original level every second day. Water sampling was performed on day 2 following the volume adjustment, such that tap water addition had a minimal impact on the measurements. After the experiments, the plants were removed completely from the solution, and the substrates entangled in the roots were cleaned. After being washed with deionized water and dried with absorbent paper, the plants were weighed with an electronic balance to obtain their fresh weight.

All experiments were conducted in a greenhouse located in the South China Botanical Garden, Guangzhou, China. The experiment was conducted from 14 November to 11 December 2018. The indoor air temperature was 14–23 °C.

### 2.4. Chemical Analysis

In this experiment, the values of T, pH, and DO were obtained using a portable multimeter (EXO2, YSI, Ohio, OH, USA); CODcr was measured using a closed reflux titrimetric method according to standard methods (GB 11914-1989); TP was determined using the ammonium molybdate spectrophotometric method (GB11893-1989); and TN was determined using a TOC analyzer (TOC-L, SHIMADZU (Hong Kong) Limited, Kyoto, Japan).

### 2.5. Statistical Analysis

All data were statistically analyzed via one-way analyses of variance using SPSS software (version 19.0; SPSS Inc., Chicago, IL, USA), and significant differences were tested using the least significant difference and Duncan multiple comparisons. Standard errors obtained from the triplicate experiments are graphically shown when they exceeded 5%.

## 3. Results

### 3.1. Variations in T, pH, and DO in Wastewater

The growth and absorption of aquatic plants are influenced by numerous environmental factors, such as solar radiation, rainfall, and temperature. The values of T, pH, and DO of the tested water with increasing hydraulic retention time (HRT) are illustrated in Figure 1.

During the experiment, the water temperature decreased significantly over time from 23 °C to 14 °C, except for a slight increase in the middle period. The pH values were between 6.5 and 8.0. The results of another study showed that no removal of the biological oxygen demand (BOD) or plant growth occurred at temperatures below 10 °C or at a pH of <5 and >10, and that the best conditions for the treatment of municipal sewage and aquatic plant growth were a temperature of 15–38 °C and pH of 7.5 [33]. Therefore, the water used in the present study was within the optimal temperature and pH ranges for macrophyte vegetative growth and aquatic microorganism development in wastewater, plant roots, and rhizomes.

In the first 8 days, the pH values of the plant groups and the control showed downward trends (Figure 1b), gradually changing from weakly alkaline conditions to weakly acidic conditions. From day 8 to day 32, the pH values of all groups kept increasing. Subsequently, the pH of group A and B went down. At the end of this experiment, the pH of group A, B, C and CK was 7.25 ± 0.05, 7.33 ± 0.11, 7.67 ± 0.33 and 8.28 ± 0.04, respectively, which of the control was significantly higher than the plant groups (*p* < 0.05). During the experiment, the pH values of each plant group first decreased and then gradually comes back up to the initial value, indicating that the water of plant groups had a certain buffering capacity. These findings also suggest that the different types of aquatic plants had different responses to pH due to their different physiological characteristics, which agree with the results of Hu et al. [34].

The initial DO concentration of the wastewater was relatively low (1.29 ± 0.02 mg/L), and increased rapidly during the first 3 days. The DO concentrations of plant groups were significantly higher than that of the control (*p* < 0.05) on day 3, and subsequently decreased before increasing (Figure 1c). At the end of the experiment, the DO concentrations of all treatments were greatly improved with respect to the initial values, with the DO concentration of the control group being significantly higher than that of the plant groups (*p* < 0.05).

### 3.2. Comparison of TN Removal in Wastewater

As a major nutrient in aquatic ecological systems, excess N can lead to the eutrophication of surface waters. Figure 2 displays the TN concentration and removal rates with increasing HRT, and shows that the initial TN concentration was 9.70 ± 1.51 mg/L. During the first 8 days, the TN concentration of each group showed a decreasing trend; the removal rates of groups A, B, C, and CK were 21.81%, 13.38%, 11.52%, and 21.36%, respectively. From day 8 to day 16, the variation trend of TN concentration in each group was not consistent. On day 27, the TN concentration of group B became higher than the initial and the removal rate reached −0.18%, which may have been related to the decay of *V. natans*. During the experiment, the TN removal rates of group B were significantly lower than those of the other groups (*p* < 0.05). At the end of the experiment, the TN removal rates of groups A, B, C, and CK were 79.10%, 46.39%, 67.46%, and 70.42%, respectively. There were no significant differences between groups A, C, and CK; however, the TN removal rate of the CK group was significantly higher than that of group B (*p* < 0.05). The average TN concentrations of groups A, B, C, and CK were 1.82 ± 0.91 mg/L, 5.03 ± 0.53 mg/L, 3.03 ± 0.16 mg/L, 2.88 ± 1.34 mg/L, respectively.

### 3.3. Comparison of TP Removal in Wastewater

The assimilation and removal of nutrients from water by aquatic plants is an efficient phytoremediation approach. Phosphorus is an important nutrient in the composition of biological life and is a highly significant limiting factor for water eutrophication.

In this experiment, the initial TP concentration was 1.81 ± 0.23 mg/L. Figure 3 shows the variation in the TP concentration and removal rate of each treatment. During the first 3 days, the TP concentration of group A decreased rapidly, and the TP removal rate reached 24.86%, which was slightly higher than that of the other groups (*p* < 0.05). At the end of the experiment, the TP removal rates increased to 89.39%, 88.37%, 91.96%, and 86.59% in groups A, B, C, and CK respectively, with no significant differences between plant groups and the CK group.

### 3.4. Comparison of COD_cr_ Removal in Wastewater

The initial COD concentration was 407.7 ± 55.0 mg/L. As shown in Figure 4, during the first 3 days, the COD concentrations of groups A and CK increased slightly, and the COD removal rates of groups A, B, C, and CK were −5.45%, 4.69%, 2.08%, and −2.5%, respectively. Subsequently, the COD concentration of all groups exhibited rapid downward trends. At the end of the experiment, the COD removal rates of groups A, B, C, and CK were 93.91%, 96.48%, 92.78%, and 87.94%, respectively, which were slightly higher than that of the CK group (p > 0.05). The average COD concentrations of groups A, B, C, and CK were 21.3 ± 7.0 mg/L, 15.53 ± 8.1 mg/L, 28.9 ± 6.16 mg/L, 52.8 ± 4.4 mg/L, respectively.

### 3.5. Visual Observations and Biomass Production

Figure 5 showed the growth status of the plants in each treatment. During the first 15 days, the biomass of the floating plants (*L. adscendens* and *I. aquatica*) increased rapidly due to the rich nutrient contents of the water. Their floating stems grew rapidly with numerous branches and roots (Figure 5b), and there were white spindle-shaped pneumatophores in clusters at the nodes of *L. adscendens*. The emergent plants (*S. chinensis* and *M. vaginalis*) floating on the water surface (Figure 5a) also developed new leaves and many roots. The height of the submerged plant *V. natans* increased, and more adventitious roots were grown for reproduction. The stems and leaves of the emergent plants *S. juncoides*, *A. calamus*, and *T. orientalis* did not grow above the ground layer, possibly because their roots were directly buried in the sand and the underground roots of the plants needed time to adapt to the new environment.

On day 16 of the experiment, considerable decay was observed in the upper leaves of *V. natans* under the floating plants. Therefore, the floating plants that blocked the light of *V. natans* were moved to other places. Over time, the decayed *V. natans* gradually decomposed in the water, and its adventitious roots grew new shoots. In the middle and late periods of the experiment, after a period of prosperous growth, the leaves of *M. vaginalis* in group A also began to wither gradually (Figure 5c,d), and some litter entered the water; the green leaves of *I. aquatica* gradually turned yellow, as the weather became colder and the nutrient content of the water decreased (Figure 5e). Figure 5e,f reveals that the water transparency of the plant groups was improved and was significantly better than that of the CK group. The net biomass growth rates of each plant in groups A, B, and C at the end of the experiment are presented in Table 2, which shows that the net biomass growth rates were not high, and were even negative in some cases. This may have been related to the growth stages of the plants in this experiment, which went from a bloom phase to a declining phase.

The amount of nitrogen absorbed by the plants and plant groups, the total removal rates of nutrients in water and absorption contribution rates of plants after 54 days treatment are shown in Table 3. In the removal of TN in water, the absorption contribution rates of plants in groups A, B, and C were 15.29%, 14.63%, and 18.24%, respectively, which were 2.05%, −2.71%, and −4.71%, respectively, in the removal of TP from water. In this experiment, all plant groups showed the phenomenon of decreasing plant biomass and releasing nitrogen or phosphorus into water. In group A, the biomass of *V. natans* decreased most obviously, by which the nitrogen and phosphorus released into the water were 95.84 mg and 15.71 mg, respectively. Although the biomass of some plants increased, it also increased the pollution load of the water body to some extent; in group A, the net biomass growth rate of *M. vaginalis* was 9.8%, but the nitrogen and phosphorus absorbed by it were −33.82 mg and −8.08 mg, respectively; in group B, the net biomass growth rate of *T. orientalis* was 37.7%, and the total phosphorus absorbed by it was −8.93 mg; in group C, the net biomass growth rate of *L. adscendens* was 19.7%, and the total phosphorus absorbed by it was −53.74 mg, which was related to the decrease of N or P concentration in plants.

## 4. Discussion

In the present study, the TN removal rates of groups A, B, C, and CK were 79.10%, 46.39%, 67.46%, and 70.42%, respectively, while the TP removal rates were 89.39%, 88.37%, 91.96%, and 86.59%, respectively. The COD removal rates of groups A, B, C, and CK were 93.91%, 96.48%, 92.78%, and 87.94%, respectively. The TN removal rate of group B was significantly lower than that of other groups (*p* < 0.05). Apart from this, there were no significant differences in the TN, TP, and COD removal rates in the wastewater of each group, which may be related to the metabolism of microorganisms in wastewater and seasonal changes of aquatic plants. In order to simulate natural eutrophic water as much as possible, the artificial wastewater in this experiment was prepared using C_6_H_12_O_6_·H_2_O, NH_4_Cl, and KH_2_PO_4_ dissolved in natural wastewater with abundant microorganisms. It was reported that the most important removal process in most treatment wetland systems were based on physical and microbial processes [35]. Vymazal [36] found that processes that affect removal and retention of nitrogen during wastewater treatment are manifold and include NH_3_ volatilization, nitrification, denitrification, nitrogen fixation, plant and microbial uptake, mineralization (ammonification), nitrate reduction to ammonium (nitrate-ammonification), anaerobic ammonia oxidation (ANAMMOX), fragmentation, sorption, desorption, burial, and leaching. Phosphorus transformations during wastewater treatment in CWs include adsorption, desorption, precipitation, dissolution, plant and microbial uptake, fragmentation, leaching, mineralization, sedimentation (peat accretion), and burial. The major phosphorus removal processes are sorption, precipitation, plant uptake (with subsequent harvest), and peat/soil accretion [11,36,37]. Many studies found that the coupling effects of microorganisms and plants play important roles in nutrient removal from wastewater [1,5,36]. However, in this experiment, the purification effect of TN, TP, and COD of the plant groups was not significantly better than that of the blank control group, which was related to the decline of plants in the cold season and the abundance of microorganisms in water.

Here, we cannot deny the ecological restoration effect of aquatic plants on the water body in the low temperature season. The most important effects of aquatic plants in relation to the wastewater treatment processes are the physical effects the plant tissues give rise to (e.g., erosion control, filtration effect, provision of surface area for attached microorganisms). The metabolism of the macrophytes (plant uptake, oxygen release, etc.) affect the treatment processes to different extents depending on the design. The macrophytes have other site-specific valuable functions, such as providing a suitable habitat for wildlife, and giving systems an aesthetic appearance [35]. The richest root system of aquatic macrophytes provided a suitable environment for aerobic microorganisms to degrade organic matter and nutrients into inorganic compounds, which were then utilized by the plants. Rhizospheric microorganisms in aquatic plants can participate in various activities, such as COD degradation and N removal and fixation.

In this experiment, the absorption contribution rates of plants in groups A, B, and C with regard to the removal of TN in water were 15.29%, 14.63%, and 18.24%, respectively, and were 2.05%, −2.71%, and −4.71%, respectively, with regard to the removal of TP from water. The absorption contribution rate of plants is affected by plant species and nitrogen and phosphorus loads in water bodies. It was reported that plant uptake of nutrients was only of quantitative importance in low-loaded systems (surface flow systems) [31,35]. Jiang [38] found that plant uptake played a major role in nitrogen and phosphorus removal in the constructed wetland treating low eutrophic water, of which the contribution rates were 46.8% and 51.0%, respectively. Due to the artificial wastewater with high nutrient load and seasonal decline of aquatic plants, the absorption contribution rates of TN in this experiment were not very high, but that of TP were very low or even negative. The result shows that the absorption of plant group has a certain effect on the removal of total nitrogen from water, while the decay of some plants increases the total phosphorus load in water.

The decay of aquatic plants has complex environmental effects on waterbodies [39]. Plant decomposition includes all the physical and chemical changes that occur after tissue senescence and death, starting with complex organic molecules and ending in simple inorganic elements [40]. The processes involved include: (1) Mechanical fragmentation by animal grazers, weathering, or other mechanisms; (2) leaching and/or autolytic production of dissolved organic matter; and (3) digestion of labile and recalcitrant materials by bacteria and fungi [41,42,43].

On day 16 of the experiment in this study, the leaves of *V. natans*, which were covered by floating plants, were fractured from the upper middle part and rotated to some extent. The net biomass growth rate of V. *natans* in groups A, B, and C was −33.1%, −27.3%, and 46.5%, respectively. By studying the decay process of six aquatic plants, Zhou et al. [44] found that floating plants and submerged plants were the most susceptible to decay, with submerged plants rotting and decomposing the most thoroughly, while emergent plants decomposing relatively slowly. Pan et al. [45] reported that *Vallisneria spiralis*—a plant of the same genus to *V. natans*—gradually began to wither and turned yellow when the water temperature was 3–10 °C in November. During that process, the authors added some nutrition to the water, which was mostly held in the plant remains. When the weather became warmer in the following year, the plants decayed more rapidly, releasing N and P to the water and sediment. It is accepted that the summer decomposition rates of plants are generally higher than those of other seasons [39]. Therefore, the present experiment was carried out during autumn in South China, when the water temperature was between 14 °C and 24 °C. Hence, the decomposed residue of *V. natans* would have been more likely to decompose even though the rate may not have been as fast as that during summer. In the first period of this experiment, the floating plants, *L. adscendens* and *I. aquatica*, both grew well due to the rich nutrient contents of the water, and their biomasses were nearly 2.5 times greater than their initial biomasses; however, at the end of the experiment, the net biomass growth rates of *L. adscendens* in groups A and C were just 80.0% and 19.7%, respectively, while that of *I. aquatica* in group B was 66.7%. These results suggest that the two floating plants were decomposed to some extent, but that decomposition started after *V. natans* began to decompose.

Plant decay may cause secondary pollution in waterbodies. The decay process and water quality effects vary depending on the plant species and the amount of residue decomposition [46]. Tang et al. [47] performed a 64-day decomposition experiment, and found that the COD and TN concentrations of water increased during the late decomposition period of *Zizania latifolia*, resulting in the deterioration of water quality. On the other hand, some studies have suggested that the existence of moderate plant residues could effectively promote the N and P cycles in waterbodies, thus reducing the N concentration to some extent and decreasing the N load [45]. In the present experiment, the tested plants absorbed nutrients from the wastewater during their growth, and their presence also improved the water environment to some extent, promoting the removal of N and P by other factors. At the same time, some plant residues entered the water and released nutrients to increase the nutrient load; hence, the N and P released by plant decay and absorption by plant growth were offset to some extent. The results indicate that a proper amount of plant residue will not lead to a significant deterioration of water quality. However, the residual biomass of aquatic plants must be controlled, fully utilized, and harvested at appropriate times to avoid excessive plant residues that can degrade water quality as temperature rises.

It has been suggested that maintaining a certain amount of aquatic plants during autumn and winter can improve the cycling process of nutrient elements such as N and P in water and prevent the deterioration of water quality [47]. In the present experiment, the litter, decay, and decomposition of plants did not cause any significant increase in nutrient concentrations of the water in the various treatments. This was related to the relatively slow decomposition rate of litter in autumn, and to the increased stability of the water due to the combination of multiple plants. Although the purification effects of the plant groups were not significantly superior to those of the CK group, there was a good purification effect on the TN, TP, and COD under the condition of plant decomposition. Hence, the introduction of appropriate aquatic plants into eutrophic waterbodies may facilitate long-term improvements in water quality.

## 5. Conclusions

In this study, all the groups had a good purification effect; the water transparency was improved in the plant groups (A, B, and C), and was significantly better than that of the CK group. The TN removal rates of groups A, B, C, and CK were 79.10%, 46.39%, 67.46%, and 70.42%, respectively, while the TP removal rates were 89.39%, 88.37%, 91.96%, and 86.59%, respectively. The COD removal rates of groups A, B, C, and CK were 93.91%, 96.48%, 92.78%, and 87.94%, respectively. Overall, the removal rates of the plant groups were slightly higher than those of the CK group, and the combination of multiple plants increased the stability of the water in groups A, B, and C. The conclusions of this study are important for the phytoremediation of wastewater in the area of south China and similar climatic zones throughout the year. However, this study was only an experiment in a relatively stable environment, and further observations of practical applications of such phytoremediation treatments are required.

## Figures and Tables

**Figure 1 ijerph-18-06162-f001:**
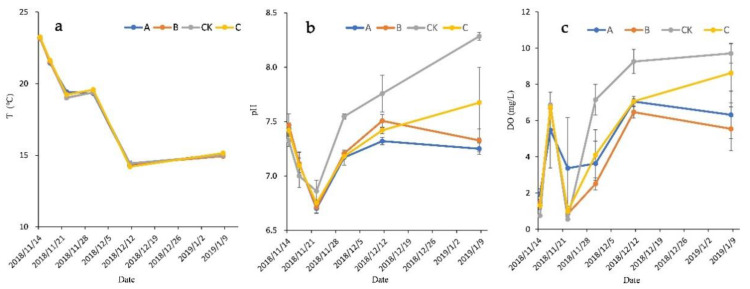
Variation in (**a**) T, (**b**) pH, and (**c**) DO of wastewater in groups: (A) *V. natans* + *L. adscendens* + *M. vaginalis* + *S. chinensis*; (B) *V. natans* + *I. aquatica* + *A. calamus* + *T. orientalis*; (C) *V. natans* + *L. adscendens* + *S. juncoides* + *T. orientalis*; (CK) control (no plants) (the same applies in the following figures).

**Figure 2 ijerph-18-06162-f002:**
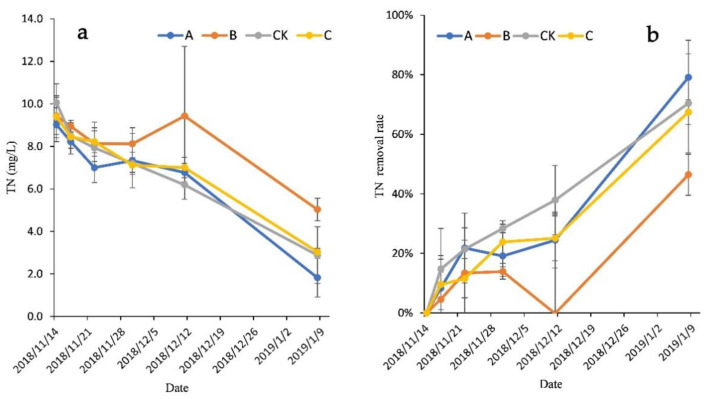
Variation in the (**a**) TN concentration and (**b**) removal rate of each treatment.

**Figure 3 ijerph-18-06162-f003:**
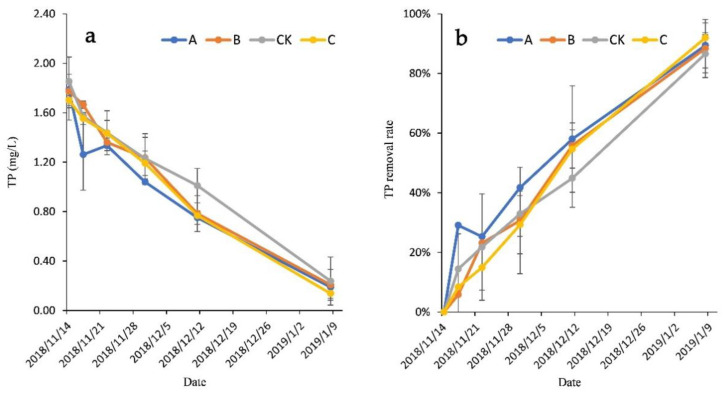
Variation in the (**a**) TP concentration and (**b**) removal rate of each treatment.

**Figure 4 ijerph-18-06162-f004:**
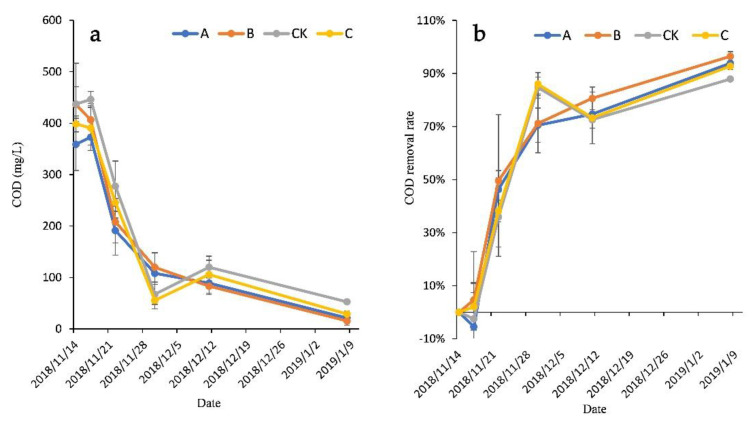
Variation in the (**a**) COD concentration and (**b**) removal rate of each treatment.

**Figure 5 ijerph-18-06162-f005:**
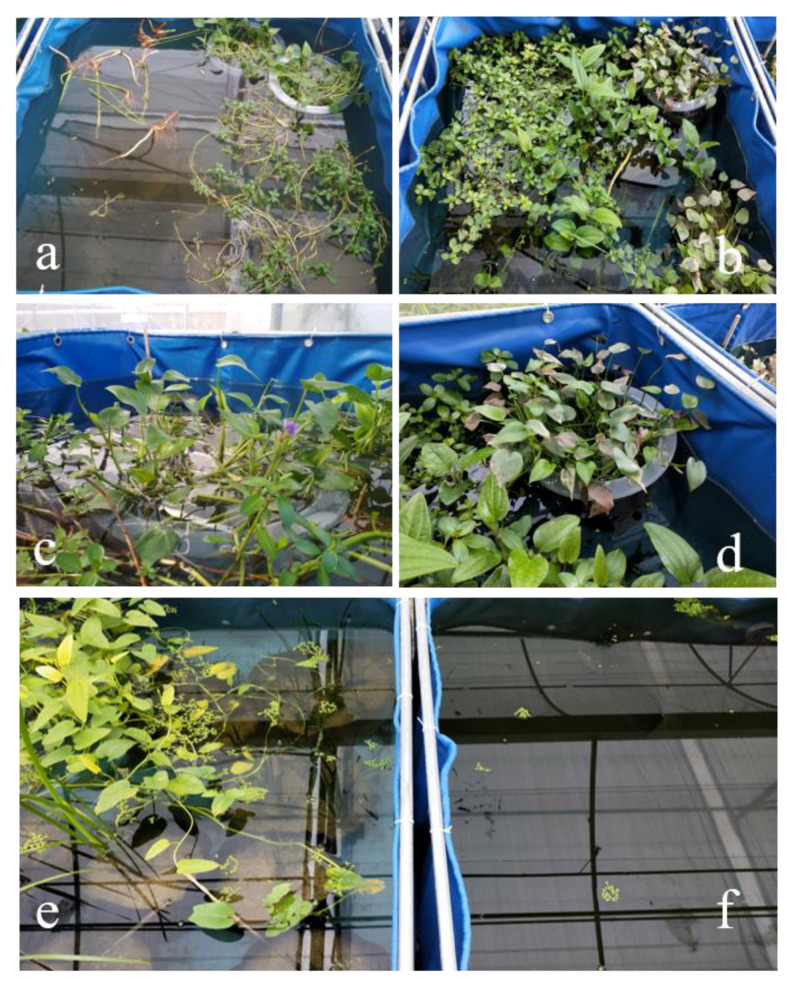
Growth status of plant combinations in wastewater: (**a**) day 0 and (**b**) day 27 of group A; (**c**) day 8 and (**d**) day 27 of *M. vaginalis*; (**e**) day 35 of group B; (**f**) day 35 of the CK group.

**Table 1 ijerph-18-06162-t001:** Aquatic plant species selected in this study.

Scientific Name	Family	Type	Group Number
*Vallisneria natans*	Hydrocharitaceae	Submerged	A, B, C
*Ludwigia adscendens*	Onagraceae	Floating	A, C
*Ipomoea aquatica*	Convolvulaceae	Floating	B
*Monochoria vaginalis*	Pontederiaceae	Floating/emergent	A
*Saururus chinensis*	Saururaceae	Floating/emergent	A
*Acorus calamus*	Araceae	Emergent	B
*Typha orientalis*	Typhaceae	Emergent	B, C
*Schoenoplectus juncoides*	Cyperaceae	Emergent	C

**Table 2 ijerph-18-06162-t002:** Biomasses of aquatic macrophytes after 54 days of treatment in artificial wastewater (g/pot, fresh weight).

Plants	Group A				Group B				Group C			
	Vn	La	Mv	Sc	Vn	Ia	Ac	To	Vn	La	Sj	To
NBG	−84.3	141	8.88	101	−13.3	40.3	−41.9	152.5	141.2	101.2	106.7	34.5
	8.86	225	−15.2	166	−86.8	181	−11.2	101.1	83.3	68.8	73.7	−26.4
	−123	114	65.0	142	−63.7	178.9	−10.3	−27.7	54.4	−51.8	106.2	−15.6
ANBG	−66.2	160	19.6	136.6	−54.6	133.4	−21.1	75.3	93.0	39.4	95.6	−2.5
NBGR	−33.1%	80.0%	9.8%	68.3%	−27.3%	66.7%	−10.6%	37.7%	46.5%	19.7%	47.8%	−1.3%

Note: NBG: net biomass growth of each plant; ANBG: average net biomass growth; NBGR: net biomass growth rate; Vn: *V. natans*; La: *L. adscendens*; Mv: *M. vaginalis*; Sc: *S. chinensis*; Ia: *I. aquatica*; Ac: *A. calamus*; To: *T. orientalis*; Sj: *S. juncoides*.

**Table 3 ijerph-18-06162-t003:** The amount of nitrogen absorbed by plants and plant groups, the total removal rates of nutrients in water and absorption contribution rates of plants after 54 days’ treatment.

Combinations	Species	NBGR	MNAP (mg)	MPAP (mg)	MNAPG (mg)	MPAPG (mg)	Removal Rate of Nutrients		Absorption Contribution Rate	
							TN	TP	TN	TP
Group A	Vn	−33.1%	−95.84	−15.71	633.87	18.84	79.10%	89.39%	15.29%	2.05%
	La	80.0%	331.50	−9.28						
	Mv	9.8%	−33.82	−8.08						
	Sc	68.3%	432.03	51.90						
Group B	Vn	−27.3%	−24.12	−6.54	370.80	−24.49	46.39%	88.37%	14.63%	−2.71%
	Ia	66.7%	166.51	18.08						
	Ac	−10.6%	54.03	−27.10						
	To	37.7%	174.38	−8.93						
Group C	Vn	46.5%	101.93	11.66	668.50	−42.42	67.46%	91.96%	18.24%	−4.71%
	La	19.7%	19.87	−53.74						
	Sj	47.8%	381.90	0.96						
	To	−1.3%	164.81	−1.31						

Note: NBGR: net biomass growth of each plant; MNAP: the mean amount of nitrogen absorbed by each plant; MPAP: the mean amount of phosphorus absorbed by each plant; MNAPG: the mean amount of nitrogen absorbed by plant groups; MPAPG: the mean amount of phosphorus absorbed by plant groups; Vn: *V. natans*; La: *L. adscendens*; Mv: *M. vaginalis*; Sc: *S. chinensis*; Ia: *I. aquatica*; Ac: *A. calamus*; To: *T. orientalis*; Sj: *S. juncoides*.

## Data Availability

The data presented in this study are available on request from the corresponding author (F.W.). The data are not publicly available due to privacy concerns.

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
