# Peer review of "Purification Efficiency of Three Combinations of Native Aquatic Macrophytes in Artificial Wastewater in Autumn"

_ijerph, 2021, doi:10.3390/ijerph18116162_

Round 1

Reviewer 1 Report

Dear authors,

thank you for resubmitting your improved manuscript. I think you added a lot of highly valuable information, especially the amount of nutrients stored within the plants is very meaningful enhancement and adds a lot to your results.

In my opinion, only your use of the term "removal rate" with respect to TN and TP stays problematic. Instead, it would be more sound to show the time series of the TN and TP concentrations. The term "removal rate" is implying that the nutrients have vanished, but it seems to me that they have only left the surface layer. A part of them is uptaken by the plants, but this seems to be a minor fraction. Therefore, it remains the question, where has the majority of it gone? I assume a lot material has sunk done, so that it stay a strong shortcoming of your study that you have not taken any water samples from the bottom. Only if you have them, you are able to compute the real removal rates. Instead, I strongly recommand to show the decrease of TN and TP concentrations (from the water surface) and speak of the decay of TN/TP concentrations instead of removal rates.

Best regards

Author Response

please check the attachment

This manuscript is a resubmission of an earlier submission. The following is a list of the peer review reports and author responses from that submission.

Round 1

Reviewer 1 Report

International Journal of Environmental Research and Public Health, Peer Review:

“Purification efficiency of three combinations of native aquatic macrophytes in eutrophic water in autumn”

General comments:

The paper submitted for review is an interesting case study of possibilities of removing nutrients using aquatic macrophytes. The authors presented experimental data resources, that allowed them to describe the changes in water chemistry under 54-days research. The results of the work are presented quite clearly, although in my opinion some aspects require clarification. In general, the work is worth publishing, but there are a few corrections that I present below.

Title

The title is not adequate to the content – You suggest, that the matrix of experiments is wastewater, not eutrophic water.

Abstract

The abstract is clearly written, stating the purpose, general methods, and results of the study. Minor note: l. 31: the abbreviation CK used here for the first time were not explained.

Introduction

The introduction provides ample background for the rest of the article. In my opinion, the review of the literature and the highlighting of the research problem is appropriate and transparent. However, the concept of the autumn research is not sufficiently explained. Since the authors considered this aspect important enough to be included in the title of the paper, they should explain the exact reason for such a research plan.

Material and Methods

l. 99: Missing info: has the sand been tested for biogenic contamination? Was there sand also in the control tanks? Was the uniformity of the experiment conditions maintained by introducing sand into all variants of the experiment?

l. 102-115, Wastewater preparation: Wastewater is not eutrophied water. You have to decide what conditions you are simulating. the nutrient levels of 10 mg/L N and 2 mg/L P are well above the definition of eutrophy. What was the reason for simulating such low water oxygenation when the experiments were foreseen under open conditions? After all, the dominant process in the first days of the experiment will be to replenish the oxygen deficiency from the atmosphere.

l. 129, The addition of tap water will always influence such an experiment. The methodology did not explain the level of the loss replenishment or the chemical composition of the tap water. This thread should be further explained, because the statement that the addition of tap water had a minimal effect on the course of the experiment is only an open assumption.

Results

l. 182-185. The description of the influence of the C vs CK combination on the oxygen conditions of the experiment does not coincide with the data in Figure 3. This graphic clearly shows that the best oxygen conditions were in the control tanks.

l. 219. The accumulation of lines on the graph does not allow for precise verification of A and CK concentrations in the third experiment because they are illegible. Perhaps reducing the thickness of the lines would improve the readability of the data.

l. 247-248. The nutrient content did not decrease only on the 16th day of the experiments, which results from the data in Figs. 4 and 5. Thus, the description is imprecise.

l. 249. Fig 4 does not describe the transparency of the water as you suggest, but nitrogen.

l. 258. Swine manure? I don't understand what your experimental matrix actually is - eutrophic water, simulated sewage or liquid manure?

Discussion

General note:

The discussion did not address the influence of the initial environmental conditions of the experiments on the quality of the tested matrix. It is clear that the driving force behind the overall changes in water chemistry over time is the general setup of the experiment. Under control conditions, the content of both nutrients decreases, and plant modifications only slightly change this scenario. This needs to be discussed - please analyze the closing effect.

I also have the impression that the discussion is a bit overloaded with an analysis of threads from the experiences of other authors and there is not enough coverage of your own experiments. Hence, some of the arguments do not refer to the course of research, but only constitute general conclusions.

Reviewer 2 Report

Dear authors,

thank you for providing this interestint manuscript dealing with the outcomes of some tank experiments, in which you studied the effects of different macrophytes on the purifiction of waste water with high nutrient amounts. While I strongly agree with your findings that a proper combination of macrophytes can play an important role to clean waste water, I have several severe problems with the setup of your experiments and the related outputs.

First of all, I undertand your results in a way that the control experiment showed nearly the same purification as the tanks with macrophytes. While you write in lines 182/183 that group C had significantly improved DO concentrations (compared to the control), it seems to me in figure 3 the other way around - the control group has the highest DO concentrations. So there seems to be a mismatch between your field results ad your shown results.

Further, your TN and TP time series are not supporting your conclussion that the macrophyte groups had a positive impact. I conclude from your figures that TN and TP concentrations have developed analog in all experiments (despite the fall back to the initial TN concentration at group B for the 5th measurement). This means the nutrient removal was as strong without the macrophytes as with the macrophytes. Here the strongest weakness of your experimental setup comes into play, as you have only analyzed the surface water concentrations (line 124). For me, it stays totally unclear where the nutrient go to during your experiments. Are the bound by phytoplankton and sank to the bottom - I don´t know from your manuscript. But how can it be that you gain so strong TN and TP removals? Where do the nutrients go? Hopefully, in the groups with macrophytes the nutrients are taken up the plants, but how can the readers know? Maybe, it all sank down to the bottom. This is not clear and must be clarified. Knowing the amount of nutrients in the whole tank would be essential to estimate realistic TN and TP removal rates, which would hopefully be enhanced due to the macrophytes.

Following this approach to study all nutrients in the whole water column, will result in a different manuscript than the one you have submitted. Therefore, I suggest to rewrite most parts of the manuscript (you may keep the introduction and most parts of the M&M section), what wouldd go beyond a revision. Therefore, I can only reject the publication in its present form, but you should think about submitting a new and improved one, following my suggestions from above.

Best regards

Reviewer 3 Report

Manuscript water-1048810 is an interesting work with a correct structure and includes clear descriptions of the study aim. 

Below some suggestions for improve the manuscript:

- There are some inaccuracies, typos, and errors that need to be resolved

- The not excellent quality of the English style needs a revision by an English colleague or a professional editor in the field.